# Pharmacokinetics of Bupivacaine Following Administration by an Ultrasound-Guided Transversus Abdominis Plane Block in Cats Undergoing Ovariohysterectomy

**DOI:** 10.3390/pharmaceutics14081548

**Published:** 2022-07-25

**Authors:** Marta Garbin, Javier Benito, Hélène L. M. Ruel, Ryota Watanabe, Beatriz P. Monteiro, Petra Cagnardi, Paulo V. Steagall

**Affiliations:** 1Faculty of Veterinary Medicine, Université de Montréal, Saint-Hyacinthe, QC J2S 2M2, Canada; marta.garbin@umontreal.ca (M.G.); javier.benito@umontreal.ca (J.B.); helene.ruel.2@umontreal.ca (H.L.M.R.); ryota.watanabe@umontreal.ca (R.W.); or pmortens@cityu.edu.hk (B.P.M.); 2Department of Veterinary Medicine and Animal Sciences, University of Milan, 26900 Lodi, Italy; petra.cagnardi@unimi.it; 3Department of Veterinary Clinical Sciences and Centre Animal Health and Welfare, Jockey Club College of Veterinary Medicine and Life Sciences, City University of Hong Kong, Hong Kong, China

**Keywords:** pharmacokinetics, feline, bupivacaine, pain, analgesia, regional anaesthesia, TAP block, bioavailability, toxicity, safety

## Abstract

Bupivacaine is commonly used for peripheral nerve block in veterinary medicine. This study described the pharmacokinetics of two doses of bupivacaine following administration by an ultrasound-guided *transversus abdominis* plane (TAP) block in cats undergoing ovariohysterectomy. Twelve healthy female adult cats were included in a randomized, prospective, blinded clinical trial. Anaesthetic protocol included acepromazine–buprenorphine–propofol–isoflurane–meloxicam. Each cat received 1 mL/kg of bupivacaine 0.2% or 0.25% (BUPI-2 and BUPI-2.5, respectively) via bilateral two-point TAP block before surgery (n = 6/group). Plasma concentrations of bupivacaine were detected using liquid chromatography-mass spectrometry. A one-compartment model and non-compartmental analysis described the pharmacokinetic parameters. Bupivacaine was detected up to 480 min (335 ± 76 in BUPI-2 and 485 ± 198 ng/mL in BUPI-2.5). For BUPI-2 and BUPI-2.5, maximum plasma concentrations were 1166 ± 511 and 1810 ± 536 ng/mL at 33 ± 14 and 47 ± 22 min, clearance was 5.3 ± 1.8 and 4.9 ± 1.5 mL/min/kg, and elimination half-life were 253 ± 55 and 217 ± 52 min, respectively. The two doses of bupivacaine via TAP block produced concentrations below toxic levels in cats. A dose of 2.5 mg/kg bupivacaine was safe to be administered using this block in healthy cats.

## 1. Introduction

The *transversus abdominis* plane (TAP) block has become a common regional anaesthetic technique for postoperative pain management in a variety of abdominal procedures in both humans and animals [1,2,3,4]. The technique generally involves the injection of a local anaesthetic in the fascia layer superficial to the *transversus abdominis* muscle, where branches of the thoracolumbar spinal nerves (from the 9th to the 12th thoracic and from the 1st to the 3rd lumbar) are located [5]. Ultrasound (US)-guided techniques are now used to perform the TAP block as the US allows identification of the anatomical landmarks and direct visualization of the needle positioning and anaesthetic spread into the fascial plane [6]. These techniques should improve the safety and efficacy of a local anaesthetic block.

In cats, a US-guided TAP block with bupivacaine has been reported for postoperative pain management as part of a multimodal approach [4]. However, studies on the safety and efficacy of this technique using bupivacaine are lacking in this species and warrant further investigation, particularly in terms of appropriate anatomical landmarks, injectate spread, potential adverse effects, drug pharmacokinetics and postoperative analgesia. In a recent cadaver and computed tomography study, our research group demonstrated that a TAP block using subcostal and lateral injection points (0.25 mL/kg/point) produced a wider injectate spread around the ventral branches of thoracolumbar spinal nerves than a single lateral in-plane injection using a higher volume (0.5 mL/kg/point) in cats [7]. Intraperitoneal injection is a possible risk as minimum injectate was found in three out of twenty-four injections through dissection and computed tomography [7].

In human medicine, studies in adults reported increased plasma concentrations of ropivacaine after TAP block that could be associated with toxicity even with doses that should normally be considered safe [8,9]. Therefore, pharmacokinetic studies of local anaesthetics are important to determine the safety profile of regional anaesthetic techniques such as the TAP block before widespread clinical application. To the authors’ knowledge, plasma concentrations and the pharmacokinetics of bupivacaine, a long-acting and popular local anaesthetic [10], have never been investigated in veterinary medicine following a TAP block. The objective of this study was to determine the pharmacokinetics of two doses of bupivacaine after administration of a US-guided TAP block using a bilateral subcostal and lateral in-plane technique in cats undergoing ovariohysterectomy (OVH). The authors hypothesized that peak plasma concentrations of bupivacaine, at the two doses used in this study, would not exceed the plasma bupivacaine concentrations reported to cause systemic toxicity in a clinical population of healthy cats [11,12].

## 2. Materials and Methods

This study was approved by the local animal care committee (*Comité d’éthique de l’utilisation des animaux*) of the Université de Montréal (21-Rech-2153) and conducted according to Canadian Council on Animal Care guidelines. Written consent for participation in the study was obtained for each patient.

### 2.1. Pharmacokinetic Study

#### 2.1.1. Animals

Twelve mixed-breed, adult female cats from a local animal shelter were admitted to the veterinary teaching hospital (*Centre hospitalier universitaire vétérinaire*) of the Faculty of Veterinary Medicine, Université de Montréal, for elective OVH. Cats were included in the study if they were deemed healthy based on physical examination and blood analysis (haematocrit and total protein) and were classified as American Society of Anesthesiologists physical status score I. Exclusion criteria included: body weight ≤2 kg and ≥6 kg; body condition score <3 and >7 on a scale from 1 to 9; pregnancy or lactation; any contraindication for locoregional anaesthesia; cardiac dysrhythmias; anaemia (hematocrit <25%); hypoproteinemia (total protein <59 g/dL); and clinical signs of disease (i.e., upper respiratory tract infection, gastrointestinal disease with diarrhea or vomiting). Cats were housed individually in stainless steel adjacent cages in a cat ward for at least 24 h before general anaesthesia. Each cage was equipped with water and food, a litter box, a blanket, a cardboard box, and a hanging toy. Cats were fed with a commercial diet twice daily and ad libitum water.

#### 2.1.2. Experimental Design and Treatment Groups

This study was a prospective, randomized, blinded clinical trial. Cats were randomized (www.randomization.org; accessed on 22 October 2021) to receive one of the following two TAP blocks using bupivacaine 2 mg/kg body weight (Bupivacaine 0.2%, BUPI-2 group, n = 6) or bupivacaine 2.5 mg/kg body weight (Bupivacaine 0.25%, BUPI-2.5 group, n = 6). For each cat, the local anaesthetic solution was prepared, calculating the assigned dose (mg/kg) and diluting the bupivacaine hydrochloride 0.5% (Bupivacaine Injection BP 0.5%; SteriMax Inc., Oakville, ON, Canada) with 0.9% saline to a volume of 1 mL/kg. Randomization and treatment preparation were performed by two veterinarians who were not involved with drug administration (P.V.S./R.W.).

#### 2.1.3. Anaesthesia and Surgery

Food but not water was withheld up to 12 h before anaesthesia. Premedication was performed with intramuscular acepromazine (0.02 mg/kg; Atravet^®^, Boehringer Ingelheim, Burlington, ON, Canada) and buprenorphine (0.02 mg/kg; Vetergesic^®^ Multidose, Champion Alstoe Animal Health, Whitby, ON, Canada). A cream of lidocaine 2.5% and prilocaine 2.5% (EMLA^®^ cream; Aspen Pharmacare Canada Inc., Oakville, ON, Canada) was applied around one of the cephalic veins after skin preparation to facilitate venous catheterization. Approximately 10 min later, a 22 G catheter was inserted aseptically into a cephalic vein for anaesthetic and fluid therapy administration.

Propofol (PropoFlo™ 28; Zoetis Canada Inc., Kirkland, QC, Canada) was administered intravenously (IV) to effect to induce general anaesthesia and allow the placement of a supraglottic airway device (V-gel^®^ ADVANCED; DocsInnovent Ltd., Hemel Hempstead, UK). General anaesthesia was maintained with isoflurane (Isoflurane USP, Fresenius Kabi, Toronto, ON, Canada) in 100% oxygen using a veterinary anaesthesia machine and a modified Mapleson D coaxial circuit (Bain circuit). A second 22 G catheter was inserted aseptically into the other cephalic vein under general anaesthesia for blood sampling and pharmacokinetic study. Cats received 0.2 mg/kg of meloxicam subcutaneously (Metacam 0.5%; Boehringer Ingelheim, Burlington, ON, USA) and were placed in dorsal recumbency over a circulating warm water blanket. Standard anaesthetic monitoring included electrocardiogram, heart rate, noninvasive blood pressure measure, pulse oximetry, pulse rate, respiratory rate, end-tidal carbon dioxide concentration, isoflurane inspiratory/expiratory fractions using a multiparametric monitor (Lifewindow™ 6000 V, Digicare Animal Health, Boynton Beach, FL, USA). Parameters were recorded every 5 min. Rectal temperature was measured before and immediately after the end of surgery. Lactated Ringer’s solution (Lactated Ringer’s injection USP, Baxter, Mississauga, ON, Canada) was administered IV at 10 mL/kg/h during anaesthesia and surgery. Anaesthesia was performed by an individual enrolled in residency training in veterinary anaesthesia and analgesia (R.W.).

An experienced veterinarian with advanced training in veterinary anaesthesia and analgesia and unaware of treatment groups (M.G.) performed all TAP injections using a 21 G, 50 mm Quincke spinal needles (BD spinal needle; Becton Dickinson & Co., Franklin Lakes, NJ, USA) connected to a prefilled syringe (Terumo syringe 5 mL; Terumo, Laguna, Philippines) by a T-port (Med-RX extension set with t-connector; CHS ltd, Oakville, ON, Canada). A linear array probe (13–6 MHz HLF-38; Sonosite Inc., Bothell, WA, USA) connected to a portable US machine (Sonosite Edge II Ultrasound System; Bothell, Sonosite Inc., Bothell, WA, USA) was used to guide the locoregional technique. The bupivacaine solution was divided into 4 aliquots of 0.25 mL/kg. Two aliquots were administered in the TAP of each hemiabdomen using a subcostal and a lateral longitudinal approach [7,13]. For the subcostal approach, the transducer was positioned parallel and immediately caudal to the costal arch, and the needle was introduced in-plane in a ventromedial-to-dorsocaudal direction until its tip was visualized into the TAP between the *rectus abdominis* and the *transversus abdominis* muscles. For the lateral approach, the transducer was positioned caudal to the last rib and parallel to the abdominal midline at the level of the axilla (3–4 cm away from the midline). The needle was introduced in-plane in a cranial-to-caudal direction until its tip was visualized in the TAP between the *transversus abdominis* and the *obliquus internus abdominis* muscles (Figure 1).

Ovariohysterectomy was performed by a veterinarian with experience in surgery (B.P.M.) approximately 5 min after the last injection of bupivacaine via TAP block. A ventral midline incision and pedicle tie technique were used as previously described [14]. The body wall and the skin were closed using simple continuous and intradermal patterns, respectively. At the end of the surgery, a 1 cm green line tattoo was applied lateral to the ventral midline incision for visual identification of a neutered animal; isoflurane was discontinued, and the cat was allowed to recover from anaesthesia.

For each cat, the following variables were recorded: the dose of propofol administered for the induction of anaesthesia; duration of surgery, defined as the time elapsed from the first incision to the last suture; duration of anaesthesia, the time elapsed from the beginning to cessation of isoflurane administration; and time to extubation, which includes the time elapsed from cessation of isoflurane administration until extubation.

### 2.2. Blood Sampling

Using a similar methodology as previous pharmacokinetic studies in our laboratory [15,16], venous blood samples (1 to 1.5 mL) were collected before (time 0) and at 5, 10, 15, 30, 60, 120, 240, 360, and 480 min after the administration of bupivacaine via TAP block. Haematocrit and total protein were re-evaluated after the last time point to exclude anaemia and hypoproteinemia. The volume of blood collected was adjusted, and less than 10% of the cat’s total blood volume was removed over the study period (1.5 mL maximum per sample). After each collection, the catheter was flushed with 1 mL of heparinized saline and the injection plug was changed so that contamination of subsequent sampling did not occur. In order to mimic clinical practice, the first five samples were collected in cats under general anaesthesia, whereas the remaining five samples were collected post-extubation in the conscious patient (one veterinarian gently restrained the cat while a second veterinarian collected the sample). Both venous catheters were removed after the final blood sampling.

Blood was placed into tubes containing potassium ethylenediaminetetraacetic acid, kept on ice for no more than 15 min and then centrifuged for 10 min at 3500× *g* (TRIAC^®^Centrifuge, Becton Dickinson and Company, Franklin Lakes, NJ, USA). Plasma was harvested and stored frozen at −70 °C until analysis.

### 2.3. Bupivacaine Analysis and Pharmacokinetics 

High-performance liquid chromatography-tandem combined with mass spectrometer was used to quantify bupivacaine in plasma. The method and validation have been described for bupivacaine in the cats [17]. Bupivacaine-free cat plasma was used to prepare the calibrators. Plasma calibration standards were prepared to enable the concentration of bupivacaine (limits of detection) from 5.00 to 5000 ng/mL. The accuracy ranged from 96.7–107.3%, and the precision observed from 1 to 4.9%.

The pharmacokinetics of bupivacaine were determined from feline plasma concentration–time data using the Phoenix WinNonLin V.8.0 software (Pharsight Corporation, St-Louis, MO, USA), which allows compartmental and non-compartmental data analyses. The one compartmental model better fit the data and provided information on terminal elimination half-life (T_1/2_; min) and clearance indexed by bioavailability (CL/F; mL/min/kg). The following pharmacokinetic variables were described by the non-compartmental analysis: the area under the plasma drug concentration–time curve from zero to the last measured time point (AUC_0-last_; min·μg/mL), the maximum concentration of drug in plasma (C_max_; ng/mL) and time to peak (T_max_; min). Lastly, the mean residence time from zero to the last measured time point (MRT_0-last_; min) was calculated using the following equation, where AUMC_0-last_ is the Area under the first moment curve from zero to the last time point measured:MRT_0-last_ = AUMC_0-last_/AUC_0-last_.

### 2.4. Statistical Analysis

This was an exploratory pharmacokinetic study, and the number of cats was determined based on previous similar reports from our laboratory [15,16] and budget constraints. Statistical analyses were carried out using SPSS Version 22.0 (IBM SPSS Statistic; IBM Corp., New York, NY, USA). Demographic data and mean pharmacokinetic parameters for each treatment group were analyzed using unpaired *t*-tests with the *p* < 0.05 set for statistical significance. All data are expressed as mean ± standard deviation (SD) unless otherwise stated.

## 3. Results

### 3.1. Animals and Procedures

Twelve adult female cats of unknown age were recruited. Body condition score, body weight, haematocrit and total protein (before and after the last sampling), the dose of propofol administered for anaesthesia induction, duration of anaesthesia and surgery, and time to extubation are shown in Table 1. None of these variables was significantly different between BUPI-2 and BUPI-2.5 groups. No perioperative complications were observed, and all cats were discharged from the hospital within 1 to 2 days after surgery.

### 3.2. Pharmacokinetic Study

There was a total of 120 samples collected during the study. The time-course of plasma bupivacaine concentrations is shown in Figure 2 and Figure 3. Individual and mean pharmacokinetic parameters are summarized in Appendix A and Table 2, respectively. The highest individual total bupivacaine peak plasma concentrations were 2080.24 ng/mL detected 30 min after injection in BUPI-2 and 2406.74 ng/mL detected 60 min after injection in BUPI-2.5. The highest individual clearance of bupivacaine was measured in BUPI-2 (8.7 mL/min/kg), whereas the lowest rate was measured in BUPI-2.5 (3 mL/min/kg). Bupivacaine was detected for up to 480 min in each cat. No signs of bupivacaine toxicosis were observed.

## 4. Discussion

The administration of two doses of bupivacaine via TAP block did not produce signs of toxicosis such as cardiovascular depression in healthy adult cats undergoing OVH. The maximum recommended dose of bupivacaine in cats is usually 1–2 mg/kg [10,18] based on toxicity studies following IV administration [11,12]. Different plasma concentrations associated with toxicity have been reported in cats [11,12,19]. In this study, peak plasma concentrations of bupivacaine following administration of 2 and 2.5 mg/kg (1.17 ± 0.51 μg/mL and 1.81 ± 0.54 μg/mL, respectively) were approximately one-third and half, respectively, of the lowest concentration reported to cause a convulsive electroencephalogram pattern in cats (3.6 ± 0.7 μg/mL) [19] and approximately one-seventh and one-fourth the concentration required to cause arrhythmias (7.5 to 10.9 μg/mL), respectively [12]. Higher plasma concentrations have been reported to cause a convulsive electroencephalogram pattern and hypotension (17 and 23 μg/mL, respectively) [12], and seizures and cardiovascular collapse in cats (37 ± 11 μg/mL and 110 ± 24 μg/mL, respectively) [11]. To the authors’ knowledge, this is the first study reporting clinical use of 2.5 mg/kg of bupivacaine in feline locoregional anaesthesia. Both doses and concentrations of bupivacaine used in the study were safe for administration by TAP block in healthy cats undergoing OVH. However, it is unclear if the high dose of bupivacaine would also be safe for other types of local anaesthetic techniques.

Systemic concentrations of local anaesthetics are not monitored in the clinical setting. Therefore, studies on the pharmacokinetics of these drugs following locoregional anaesthesia are essential to understanding the safety and dosing of these techniques in clinical practice. Our results are clinically important because locoregional anaesthesia has been recommended to improve postoperative analgesia as part of a multimodal analgesic approach and opioid-sparing anaesthetic techniques [18,20]. In addition, local anaesthetics have anti-inflammatory, anti-thrombotic, anti-microbial, and neuroprotective properties [21,22]. The efficacy of a bilateral TAP block for postoperative analgesia has been demonstrated in cats undergoing ovariectomy [4]. Therefore, the TAP block with bupivacaine might provide analgesia following different abdominal surgical procedures, but this hypothesis should be corroborated with a prospective, randomized study investigating the analgesic efficacy of this technique. A dose-dependent effect of local anaesthetics on total analgesic consumption and time to analgesic requirement was shown in children receiving bilateral TAP block [23]. Hence, a long-lasting analgesic effect is expected when a high dose of local anaesthetic is administered. Significant differences were not observed between the pharmacokinetic parameters of the two doses of bupivacaine employed in this study. However, the authors speculate that a dose of 2.5 mg/kg of bupivacaine could present a longer analgesic effect than a dose of 2 mg/kg. Further pharmacokinetic and pharmacodynamic modelling using plasma concentrations and outcomes related to clinical analgesia may provide a better understanding of dose-dependent effects of bupivacaine after locoregional administration in cats.

A consensus on dosage regimens, including volumes of administration with local anaesthetics has neither been defined in humans nor in veterinary medicine for the TAP block. Generally, the TAP block—like any other fascial plane blocks—requires a large volume of local anaesthetic solution which may lead to increased plasma concentrations associated with local anaesthetic toxicity. Local anaesthetics block sodium channels and cessation of action potential propagation [24]. Systemic absorption of local anaesthetics can depend on the dose and spread of the injected volume, tissue vascularity, and the patient’s physiological state [25]. Because local anaesthetic binds to plasma proteins—primarily *α*1-acid glycoprotein and albumin—changes in plasma protein concentrations may also influence the pharmacokinetics of a local anaesthetic and are affected by surgical trauma and systemic disease (cardiac, renal, and liver disease). Therefore, these factors should be taken into consideration in the choice of local anaesthetic dose, especially in the presence of patient co-morbidities and diseases that could lead to toxicosis.

Plasma concentrations of bupivacaine have been reported using other local anaesthetic techniques in cats [15,16,26]. In particular, the pharmacokinetics of a dose of 2 mg/kg of bupivacaine was determined following intraperitoneal (IP) instillation in cats undergoing OVH [15]. The pharmacokinetics of bupivacaine after IP and TAP administration resulted in a similar peak of plasma level (1030 ± 497.5 ng/mL vs. 1166 ± 511 ng/mL, respectively). In both studies, plasma levels of bupivacaine increased slowly, with peak plasma concentrations occurring at 30 min post-administration (30 ± 24 min by TAP block vs. 33 ± 14 min by IP) in most cats. Clearance and elimination half-life were also alike (5.8 ± 3.0 mL/min/kg and 4.78 ± 2.70 h vs. 5.3 ± 1.8 mL/min/kg and 4.22 ± 0.92 h following IP and TAP administration, respectively). It would be interesting to assess whether the two locoregional techniques provide comparable post-operative pain relief in cats or even if they could be used in combination in future prospective clinical trials. Additionally, the pharmacokinetics of bupivacaine in combination with dexmedetomidine or epinephrine were different from bupivacaine alone after IP administration [16]. It would be interesting to know if the safety and analgesic efficacy of the TAP block could be improved by the addition of adjuvant drugs such as dexmedetomidine or epinephrine as these drugs may delay systemic absorption and prolong analgesia [16]. Finally, it is not known whether pharmacokinetic parameters would be significantly changed using a different US-guided TAP technique.

Pharmacokinetic parameters were not significantly different between the two doses of bupivacaine employed in this study. However, some of the results may be of clinical relevance as, overall, the high dose of bupivacaine produced increased plasma concentrations later when compared with the low dose indicating that the drug continued to be absorbed long after the block was performed with the high dose. Sustained concentrations of bupivacaine (i.e., approximately one-fourth of the maximum concentrations) were still observed at the last time point (i.e., 480 min) in both groups suggesting that the TAP block could have a long-lasting analgesic effect in feline practice. Clearance indexed by bioavailability and terminal elimination half-life were similar between doses due to the presence of outliers. For example, one cat in group BUPI-2.5 presented clearance of 3.00 mL/min/kg and terminal half-life of 280 min. On the other hand, one cat in group BUPI-2 had clearance of 8.71 mL/min/kg and a terminal half-life of 264 min. In addition, bupivacaine terminal half-life was as long as 342 min in one cat in BUPI-2 and as short as 146 min in a cat in BUPI-2.5.

The current study presents some limitations. Firstly, active metabolites of bupivacaine were not analyzed; however, the clinical relevance of these metabolites in cats is unknown. Secondly, the plasma concentrations of bupivacaine were analyzed up to eight h after TAP administration. These concentrations did not return to pre-injection values by that last time point. Despite this could have influenced the pharmacokinetics to detect the source of variability, the slope of the curve of the natural logarithm of concentration versus time had a homogenous elimination phase without evidence of drug redistribution that could have influenced the pharmacokinetic profile. More accurate determinations could be achieved with a longer sampling time during the post-injection period (i.e., at 12 and 24 h). Thirdly, these pharmacokinetic parameters were calculated using a small sample size (n = 6/group) with high individual variability; it would be interesting to collect samples in a large sample size, potentially allowing population pharmacokinetics. Fourthly, the efficacy of the TAP block requires further investigation using a large prospective, randomized, masked, controlled clinical trial with an appropriate sample size, which is currently being performed in our laboratory. Finally, this study involved a population of healthy cats undergoing OVH. It is unknown how results could be extrapolated to cats with severe illness undergoing abdominal laparotomy and presenting hypovolemia and hypotension that could affect drug absorption, distribution, metabolism, and excretion.

## 5. Conclusions

The two doses of bupivacaine administered via TAP block produced concentrations below the toxic levels reported in cats. A dose of 2.5 mg/kg appears safe to be administered in healthy cats. Further studies are warranted to assess the efficacy of bilateral TAP block with 2.5 mg/kg bupivacaine to minimize post-surgical abdominal pain in cats.

## Figures and Tables

**Figure 1 pharmaceutics-14-01548-f001:**
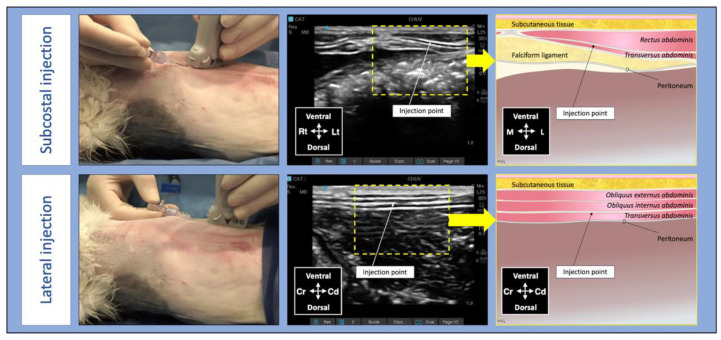
Techniques, ultrasound images and anatomical landmarks for the *transversus abdominis* plane block using lateral and subcostal injection points in a cat; Cd, caudal; Cr, cranial, L, lateral; Lt, left; M, medial; Rt, right [7].

**Figure 2 pharmaceutics-14-01548-f002:**
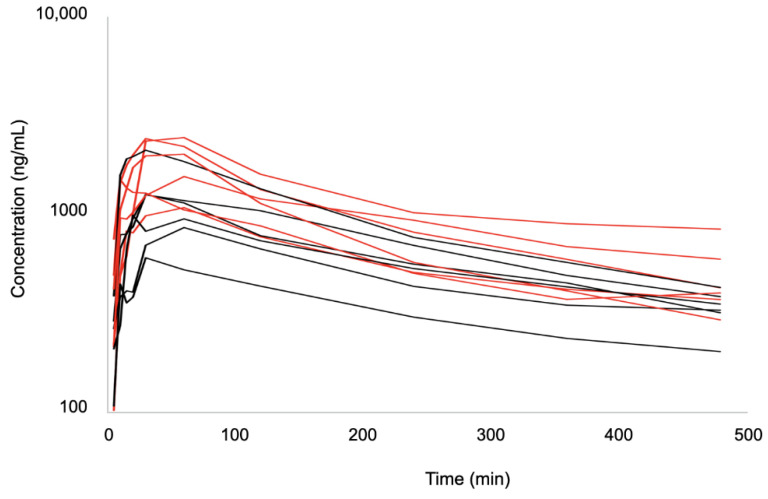
Spaghetti plot of the individual plasma concentrations of bupivacaine after injection of 2 mg/kg (BUPI-2; black) and 2.5 mg/kg (BUPI-2.5; red) by bilateral ultrasound-guided *transversus abdominis* plane block in twelve female cats (n = 6/group).

**Figure 3 pharmaceutics-14-01548-f003:**
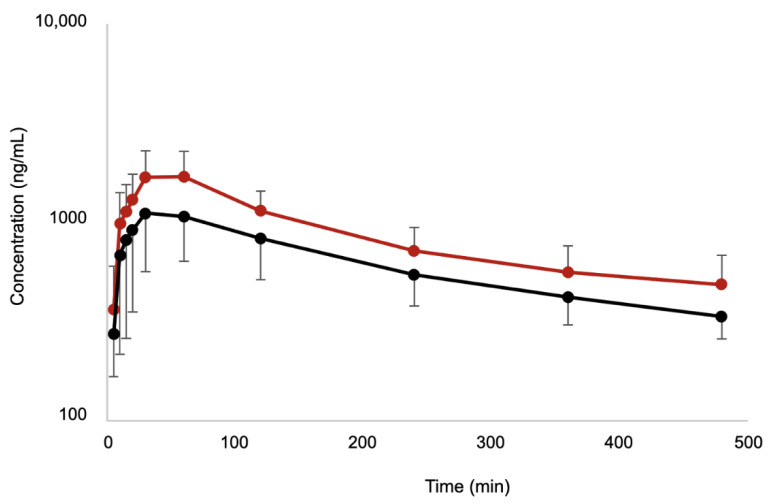
Mean plasma concentrations of bupivacaine and standard deviation after injection of 2 mg/kg (BUPI-2; black) and 2.5 mg/kg (BUPI-2.5; red) by bilateral ultrasound-guided *transversus abdominis* plane block in twelve female cats (n = 6/group).

**Table 1 pharmaceutics-14-01548-t001:** Body weight, body condition scores (BCS), haematocrit, total protein, the dose of propofol for anaesthesia induction, duration of anaesthesia and surgery, and time to extubation in cats undergoing ovariohysterectomy after administration of bupivacaine 2 mg/kg (BUPI-2) or 2.5 mg/kg (BUPI-2.5) by bilateral ultrasound-guided *transversus abdominis* plane block.

Variables	Population n = 12	BUPI-2 n = 6	BUPI-2.5 n = 6	*p*-Value
Body weight (kg)	3.68 ± 0.68	3.61 ± 0.69	3.75 ± 0.72	*p* = 0.74
BCS (1 to 9)	4 ± 1	4 ± 1	4 ± 1	*p* = 1.00
Haematocrit _before_ (%) *	42.9 ± 4.7	43.0 ± 4.9	42.8 ± 4.9	*p* = 0.95
Haematocrit _after_ (%) *	35.9 ± 4.4	34.7 ± 5.5	37.1 ± 3.1	*p* = 0.36
Total protein _before_ (g/L) *	65.3 ± 6.00	62.3 ± 5.3	68.3 ± 5.3	*p* = 0.08
Total protein _after_ (g/L) *	58.3 ± 6.1	57.5 ± 8.1	58.2 ± 3.8	*p* = 0.86
Dose of propofol (mg/kg)	4.9 ± 0.7	4.8 ± 0.7	4.9 ± 0.8	*p* = 0.79
Duration of anaesthesia (min)	38.9 ± 8.1	39.0 ± 9.5	38.8 ± 7.3	*p* = 0.97
Duration of surgery (min)	17.6 ± 5.4	17.8 ± 7.5	17.3 ± 3.0	*p* = 0.89
Time to extubation (min)	3.25 ± 1.86	2.50 ± 1.05	4.00 ± 2.28	*p* = 0.19

* Haematocrit normal range 28 to 47%; Total protein normal range 59 to 81 g/L.

**Table 2 pharmaceutics-14-01548-t002:** Mean ± SD pharmacokinetic data for bupivacaine after bilateral *transversus abdominis* plane injection of 2 mg/kg or 2.5 mg/kg (BUPI-2 and BUPI-2.5, respectively) in twelve cats (n = 6/group).

Parameters	Units	BUPI-2 n = 6	BUPI-2.5 n = 6	*p*-Value
C_max_	ng/mL	1166 ± 511	1810 ± 536	*p* = 0.06
T_max_	min	33 ± 14	47 ± 22	*p* = 0.22
CL/F	mL/min/kg	5.3 ± 1.8	4.9 ± 1.5	*p* = 0.68
T_1/2_	min	253 ± 55	217 ± 52	*p* = 0.27
AUC_0-last_	Min·µg/mL	297 ± 104	418 ± 115	*p* = 0.09
AUMC_0-last_	min·min·µg/mL	56,214 ± 16,681	77,141 ± 24,478	*p* = 0.12
MRT_0-last_	min	192 ± 11	184 ± 18	*p* = 0.38
C_480_	ng/mL	335 ± 76	485 ± 198	*p* = 0.13

C_max_—Maximum bupivacaine plasma concentrations; T_max_—Time to maximum concentrations; CL/F—Relative clearance indexed by bioavailability; T_1/2_—Terminal elimination half-life; AUC_0-last_—Area under the plasma concentration-time curve from zero to the last time point measured; AUMC_0-last_—Area under the first moment curve from zero to the last time point measured; MRT_0-last_—Mean residence time from zero to the last time point measured; C_480_—Plasma concentrations of bupivacaine at 480 min.

## Data Availability

Data are available from authors upon reasonable request.

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
