# Peer review of "Pharmacokinetics of Bupivacaine Following Administration by an Ultrasound-Guided Transversus Abdominis Plane Block in Cats Undergoing Ovariohysterectomy"

_pharmaceutics, 2022, doi:10.3390/pharmaceutics14081548_

Round 1

Reviewer 1 Report

A really interesting and clinically applicable research

Author Response

Thank you for your comments

Reviewer 2 Report

I agree with the authors, especially with Ultrasound (US)-guided techniques.  “The transversus abdominis plane (TAP) block has become a common regional anesthetic technique for postoperative pain management in a variety of abdominal procedures in both humans and animals.”  Animal models of TAP are also valuable in the research and development of local anesthetics. 

[About the Concentration of bupivacaine]

In “Abstract”,

(Line 20-22) “Each cat received 1 mL/kg of bupivacaine 2% or 2.5% (BUPI-2 and BUPI-2.5, respectively) via bilateral two-point TAP block before surgery (n = 6/group)”.

In “2.1.2. Experimental Design and Treatment Groups”,

(Line 92-96) “Cats were randomized (www.randomization.org; accessed October 22nd, 2021), to receive one of the following two TAP blocks using bupivacaine 2 mg/kg body weight (Bupivacaine 2%, BUPI-2 group, n = 6) or bupivacaine 2.5 mg/kg body weight (Bupivacaine 2.5%, BUPI-2.5 group n = 6).”

and

(Line 96-98) “For each cat, the local anesthetic solution was prepared calculating the assigned dose (mg/kg) and diluting the bupivacaine hydrochloride 0.5% (Bupivacaine Injection BP 0.5%; SteriMax Inc., Oakville, ON, Canada) with 0.9% saline to a volume of 1 mL/kg.”

Market preparation of bupivacaine is prepared at 0.5%, so how do the authors make a 2% or 2.5% concentration?

In the “U.S. FDA Drugs Database”, the highest preparation concentration of bupivacaine is 0.5%.

In Line 97-98, “Bupivacaine Injection BP 0.5%; SteriMax Inc., Oakville, ON, Canada”.

The authors need an explanation of this.

Further, if bubivacaine is 2% or 2.5%, the authors should provide the evidence of no local tissue injury.

[About Blood Sampling]

(Line 163-165) Using similar methodology as previous pharmacokinetic studies in our laboratory, venous blood samples (1 to 1.5 mL) were collected before (time 0) and at 5, 10, 15, 30, 60, 120, 240, 360 and 480 min after the administration of bupivacaine via TAP block. There are 10 sampling points.

(Line 171-174) “In order to mimic clinical practice, the first five samples were collected in cats under general anesthesia, whereas the remaining six samples were collected post-extubation in the conscious patient (one veterinarian gently restrained the cat while a second veterinarian collected the sample).” There are 11 sampling points.

Slip of the pen?

Or are the original sampling points different?

[About the Parameters of PK]

In Table 2,

Why is the Tmax of BUPI-2 (33 ± 14 min) faster than that of BUPI-2.5 (47 ± 22 min)?

In Table S1 of “pharmaceutics-1782613-supplementary”,

Why is the Cmax very variable in both groups?

The authors need to revisit the Figure 2.

Author Response

I agree with the authors, especially with Ultrasound (US)-guided techniques.  “The transversus abdominis plane (TAP) block has become a common regional anesthetic technique for postoperative pain management in a variety of abdominal procedures in both humans and animals.”  Animal models of TAP are also valuable in the research and development of local anesthetics. 

[About the Concentration of bupivacaine]

In “Abstract”,

(Line 20-22) “Each cat received 1 mL/kg of bupivacaine 2% or 2.5% (BUPI-2 and BUPI-2.5, respectively) via bilateral two-point TAP block before surgery (n = 6/group)”.

In “2.1.2. Experimental Design and Treatment Groups”,

(Line 92-96) “Cats were randomized (www.randomization.org; accessed October 22nd, 2021), to receive one of the following two TAP blocks using bupivacaine 2 mg/kg body weight (Bupivacaine 2%, BUPI-2 group, n = 6) or bupivacaine 2.5 mg/kg body weight (Bupivacaine 2.5%, BUPI-2.5 group n = 6).”

and

(Line 96-98) “For each cat, the local anesthetic solution was prepared calculating the assigned dose (mg/kg) and diluting the bupivacaine hydrochloride 0.5% (Bupivacaine Injection BP 0.5%; SteriMax Inc., Oakville, ON, Canada) with 0.9% saline to a volume of 1 mL/kg.”

Market preparation of bupivacaine is prepared at 0.5%, so how do the authors make a 2% or 2.5% concentration?

In the “U.S. FDA Drugs Database”, the highest preparation concentration of bupivacaine is 0.5%.

In Line 97-98, “Bupivacaine Injection BP 0.5%; SteriMax Inc., Oakville, ON, Canada”.

The authors need an explanation of this.

Further, if bubivacaine is 2% or 2.5%, the authors should provide the evidence of no local tissue injury.

Answer: Thank you for your comments.

Bupivacaine 0.2% or 0.25% was administered via TAP block. This concentration was obtained by diluting bupivacaine hydrochloride 0.5% with 0.9% saline. We corrected the bupivacaine concentrations throughout the manuscript. Lines 95-97 state: ‘For each cat, the local anesthetic solution was prepared calculating the assigned dose (mg/kg) and diluting the bupivacaine hydrochloride 0.5% (Bupivacaine Injection BP 0.5%; SteriMax Inc., Oakville, ON, Canada) with 0.9% saline to a volume of 1 mL/kg”.

On a different note, bupivacaine injection BP was used at concentrations of 0.2% and 0.25%, and not 2% or 2.5%. Indeed, concentrations of bupivacaine 0.25% are commercially available and used in Canada. Both concentrations of bupivacaine (0.25% and 0.5%) are widely used in both veterinary and human medicine without any complications when appropriate techniques are used for loco-regional anesthesia.

[About Blood Sampling]

(Line 163-165) “Using similar methodology as previous pharmacokinetic studies in our laboratory, venous blood samples (1 to 1.5 mL) were collected before (time 0) and at 5, 10, 15, 30, 60, 120, 240, 360 and 480 min after the administration of bupivacaine via TAP block.” There are 10 sampling points.

(Line 171-174) “In order to mimic clinical practice, the first five samples were collected in cats under general anesthesia, whereas the remaining six samples were collected post-extubation in the conscious patient (one veterinarian gently restrained the cat while a second veterinarian collected the sample).” There are 11 sampling points.

Slip of the pen?

Or are the original sampling points different?

Answer: A total of 10 blood samples were collected. The sentence in lines 171-174 was modified accordingly. Thank you for pointing this out.

[About the Parameters of PK]

In Table 2,

Why is the Tmax of BUPI-2 (33 ± 14 min) faster than that of BUPI-2.5 (47 ± 22 min)?

Answer: Similar investigations on the PK of local anesthetics administered by TAP block indicated that plasma concentrations increase more slowly than other regional blocks. Therefore, this could be the reason why Tmax was reached later in BUPI-2.5 than BUPI-2: bupivacaine was slowly absorbed from the TAP using BUPI-2.5 when compared with BUPI-2.

To the authors’ knowledge, there are no investigations in the literature comparing the PK parameters of two concentrations of bupivacaine after a TAP block. However, in a recent study on TAP block in children, Tmax was achieved later in a group receiving the low-volume-high-concentration of levobupivacaine than the group high-volume-low-concentration (not significant difference; doi: 10.1213/ANE.0000000000003736). Additionally, large data variability was observed in this study and results were not significantly different between treatments, except for Cmax that has a p value close to it (0.06). As state in the limitations of our study, a larger group of animals would help in clarifying these results.

In Table S1 of “pharmaceutics-1782613-supplementary”,

Why is the Cmax very variable in both groups?

Answer: Individual variability was probably the cause for variable Cmax values.

Other studies reported variable Cmax after TAP drug administration. For example, El Sherif et al. (2022) analyzed the PK of dexmedetomidine by TAP block and reported similar subject variability in the cohort (https://doi.org/10.2147/JPR.S335806). In the investigation conducted by Sola et al. (2019), a similar subject variability was observed in Cmax after the administration of levobupivacaine by TAP block (doi: 10.1213/ANE.0000000000003736).

The authors need to revisit the Figure 2.

Answer: Figure 2 has now similar units, size and scale as Figure 3. Figure 2 explains the large variability observed in the time-concentration plot of each cat. This gives a more clear and rapid observation of individual variability.